# Effects of Supplement Irrigation and Nitrogen Application Levels on Soil Carbon–Nitrogen Content and Yield of One-Year Double Cropping Maize in Subtropical Region

Gui-Yang Wang [†], Yu-Xin Hu [†], Yong-Xin Liu, Shakeel Ahmad [ID] and Xun-Bo Zhou *[ID]

Guangxi Colleges and Universities Key Laboratory of Crop Cultivation and Tillage, Agricultural College of Guangxi University, Nanning 530004, China; wgylonger@163.com (G.-Y.W.); hyxjyo@163.com (Y.-X.H.); yosinl@foxmail.com (Y.-X.L.); shakeel1287@hotmail.com (S.A.)
* Correspondence: xunbozhou@gxu.edu.cn
† Indicates equal author contributions.

**Abstract:** Inappropriate irrigation conditions and nitrogen application can negatively affect soil carbon–nitrogen content and yield of maize, as well as can lead to underground water pollution and soil degradation. A two year (2018, 2019) field experiment was carried out to determine the effect of irrigation and N, alone and in combination on maize grain yield, grain nitrogen content, soil inorganic N and MBC of one-year double cropping maize (*Zea mays* L.) in a subtropical region. Split plot design was adopted, with main plots consisting of two water regimes: drip irrigation (drip irrigation to keep soil water content no less than 70% of maximum field capacity) and rainfed (no irrigation during growing period). Split-plot treatments consisted of five nitrogen application levels, including 0 (N0), 150 (N150), 200 (N200), 250 (N250), and 300 kg/ha (N300). The results of two-year field experiment showed that soil irrigation nitrogen interaction had a significant influence on the all measured parameters. In detail, soil $NH_4^+$-N and $NO_3^-$-N content, total nitrogen (TN), soil organic carbon (SOC) and grain nitrogen contents under the combined treatment of N250 and supplementary irrigation were higher relative to other treatments. Compared with rainfed, maize yield, thousand grains weight (TGW) and harvest index increased by 22.0%, 7.7%, and 15.2% under supplemental irrigation. Yield and TGW N300 were 287 kg/ha and 3.1 g higher than those of N250, and yield and TGW of N250 were 59.4% and 23.1% higher than those of N0, respectively. The yield of spring maize was 24.0% significantly higher than that of autumn maize. Therefore, we suggested that 250 kg/ha nitrogen application fertilizer combined with supplementary irrigation can improve soil fertility and annual maize yield in subtropical one-year double cropping region.

**Keywords:** nitrogen; drip irrigation; maize; soil carbon–nitrogen; yield



## 1. Introduction

In the current food production scenario across major cropping systems of the world, crop yield is affected more by available resources of water and nitrogen (N), rather than by crop genetics [1]. There is increasing demand for food, but the water resources are depleting, and environmental pollution is increasing day by day which has further necessitated the optimizations and management of the water resources for the crops. It is obvious that suitable irrigation and nitrogen application have enhanced the yield, which is an important factor for efficient absorption and utilization of N by the plants resulting in higher yield, especially higher dose of N has a dominant effect on crop yield [2–6]. However, in order to increase yields, water and nitrogen are often overused, resulting in waste of water resources and accumulation of inorganic nitrogen in the soil. There has been growing concern in the research community over the intensive use of inorganic N, due to the potential threat to the environment. Ahmed et al. [7] and Hao et al. [8] mentioned that excessive nitrogen fertilizer application affects environment, while contributing to global

warming, soil acidification, and water eutrophication. It also reduces fertilizer utilization efficiency and causes environmental pollution and ecological damage [9,10]. Change in global climate increases the evaporation of water and directly affects regional precipitation imbalance [11]. Therefore, it is very vital to optimize the interactive utilization of water and N to achieve the maximum crop yield.

Water shortage and soil impoverishment are the two main factors affecting crop yield. Fertile soil with improved chemical and physical properties can be achieved with fertilization and the effective utilization of applied fertilizer by plants can be promoted by irrigation practices, which is ultimately enhancing the water use efficiency and crop yield [12–14]. Furthermore, maintaining a certain moisture field capacity can increase organic carbon and soil moisture content of the 40 cm soil layer, while water content affects mineralization of soil organic nitrogen [15,16]. Soil organic matter and total nitrogen (TN) reflect supply of soil N; application of fertilizers is beneficial to increase organic carbon and TN contents in soil with low organic matter content [17,18].

Most of N content in soil is in organic combined form, and inorganic N generally accounts for 1–5% of TN. Ease and decrease in soil organic matter and N are related to strength of bioaccumulation and decomposition, environment, farming measures, and other factors, especially water and temperature conditions [19,20]. Long-term application of N, phosphorus (P), potassium (K), and organic fertilizers can increase soil microbial biomass carbon (MBC) and N [21,22]. The MBC is an important source of available nutrients for plants. Soil carbon pool is an important fertility index of farmland ecosystems. Fertilization significantly affects soil physical and chemical properties and microbial community [23,24].

Guangxi is a typical subtropical monsoon humid region with abundant rainfall, but its spatial and temporal distribution is uneven. Maize is the top food crop in the region growing in a double cropping system (spring and autumn maize). It is also true that maize crop is highly dependent on water and N for the optimum yield. The unbalanced application of these two negatively affects NUE, WUE and reduces crop yield while enhancing soil acidification [25–27]. Furthermore, the lack of reasonable water resource management and appropriate water treatment technologies leads to freshwater resource and water quality deterioration [28,29]. Agricultural ecological environment problems need to be alleviated by reducing irrigation water consumption and improving water use efficiency [25,30,31]. Therefore, knowledge of target environments is essential when developing strategies to increase the grain yield of maize under water shortage [32].

Previous studies showed either alone or combined effects of irrigation and nitrogen application on different crop species and on transport and distribution of soil carbon–nitrogen (C–N) content using double cropping system [8]. Although one-year double maize has better potential to increase productivity, most research focused on one-year maize. Additionally, some studies only consisted of pot experiments conducted under the control environment. However, there is lack of knowledge regarding the interactive influence of supplementary irrigation and nitrogen application on maize crop in actual farmer field condition. Therefore, this research was focused on (1) determining the soil C–N transfer and distribution, soil microbial content, total plant nitrogen and yield component; (2) clarifying the relationship between soil C–N, plant N, and maize yield under rainfed and supplementary irrigation combined with nitrogen fertilizer application; (3) to some extent, obtaining a good agricultural measure for saving water and fertilizer of one-year double cropping maize in a subtropical region.

## 2. Material and Methods

### 2.1. Site Description

The experiment was conducted at Agronomy Farm of Guangxi University, Nan'ning, Guangxi in China (22°50′ N, 108°17′ E) from March 2018 to December 2019. This region is characterized by a subtropical climate, with annual average air temperature of 21.6 °C from 2018 to 2019. Amounts of rainfall and irrigation from March to December were 1028.2 and 86.9 mm (2018), 870.8 mm and 189.4 mm (2019), respectively (Figure 1). The soil is clay by

international standard for soil texture classification, and physical and chemical properties of 0–20 cm soil layer are as follows: soil bulk density 1.50 g/cm$^3$, pH 5.4, available N 126.2 mg/kg, available P 40.0 mg/kg, available K 124.5 mg/kg, soil organic matter content 17.5 g/kg, and field capacity 37.2% (*v/v*).

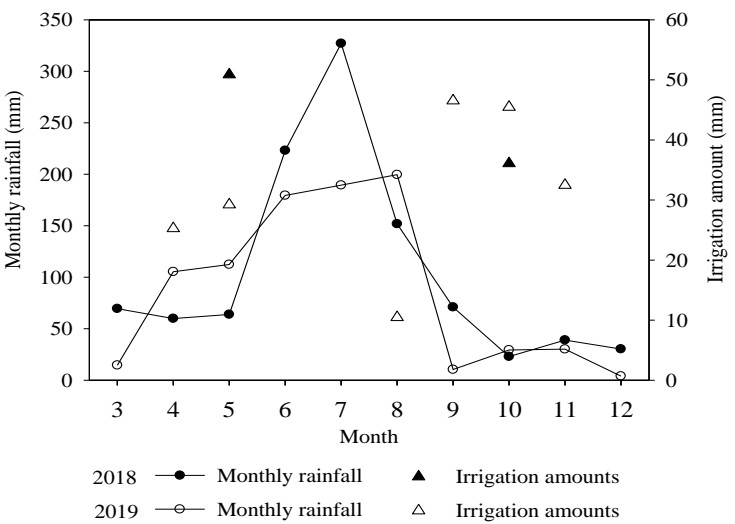

**Figure 1.** Monthly rainfall and irrigation amounts in 2018–2019.

## 2.2. Experimental Design

The experiment was conducted in a split plot design, with the main plots under two water conditions: rainfed (natural rainfall) and supplementary irrigation (drip irrigation was adopted when soil volume moisture content was lower than 70% of maximum water-holding capacity of field). Split plots were treated with five nitrogen application levels, such as, 0 (N0), 150 (N150), 200 (N200), 250 (N250), and 300 kg/ha (N300). Soil water content was measured regularly by TDR-100 (Spectrum Technologies Inc., Aurora, IL, USA) to maintain the required field moisture capacity. The area of experimental plot was 4.2 × 3.9 m, with three replicates.

Each plot was supplied with 120 kg/ha of $P_2O_5$ and $K_2O$ each before sowing as base fertilizers. Nitrogen fertilizer was divided into base fertilizer (50%) and topdressing at large trumpet period (50%). The N, P, and K fertilizers used were urea (N content 46.4%), calcium magnesium phosphate ($P_2O_5$ content 18%), and KCl ($K_2O$ content 60%). Planting density of maize (Wanchuan 1306) was 55,555 plants/ha, with 0.6 m row spacing and 0.3 m plant spacing. All other agronomic variables remained constant between the treatments (i.e., crops, tillage regime, pesticides, harvesting, etc.). Spring maize was sown on 22 March 2018 and 27 February 2019 and harvested on 12 July 2018 and 7 July 2019, respectively. While, autumn maize was sown on 11 August 2018 and 10 August 2019 and harvested on 16 December 2018 and 21 November 2019, respectively.

## 2.3. Determination Items and Methods

### 2.3.1. Plant Total Nitrogen (TN) Content

The TN content of plants was measured at flowering and maturity stages according to method of Li et al. [20]. For example, stems, leaves, and ears samples were collected and dried at flowering stage and stem, leaves, ears and grain samples were collected and dried at maturity stage. The dried samples were crushed and passed through 100 mesh (0.149 mm) sieve. Equal amount of 0.0001 g sample was digested with $H_2SO_4$–$H_2O_2$ and diluted to 50 mL using distilled water. The total N was measured by Auto Analyzer 3 Continuous Flow Chemical Analyzer (Bran+Luebbe, Norderstedt, Germany).

### 2.3.2. Soil N and C Contents

The sampling collection was done by 5-point method before harvest of maize crop. The soil was sieved through a 2 mm mesh sieve and divided into two parts. One part of fresh soil was stored in a refrigerator at 4 °C to determine soil MBC, $NO_3^--N$, and $NH_4^+-N$. The other soil sample was naturally air dried, ground using quarter method, and then passed through a 100 mesh sieve to determine soil TN and organic carbon (SOC).

Chloroform fumigation–$K_2SO_4$ extraction method was used to measure MBC [33]. In detail, 15 g fresh soil sample were spread into culture dish, then placed into a drying tank, and incubated in dark for 7 days under vacuum. This process was repeated six times. Three soil samples were extracted by titration, and another three soil samples were chloroform extracted to fumigate for 24 h before extraction and titration.

$$MBC = EC/KEC \tag{1}$$

where EC and KEC represent difference value and conversion coefficient (0.38) between fumigated and nonfumigated soils, respectively.

The SOC was determined by potassium dichromate volumetric and external heating methods [34].

A total of 5 g fresh soil were placed in a 50 mL white bottle by adding 25 mL of 1 mol/L KCl solution and shaken for 30 min (200 rpm, 28 °C) on ZWYR-4912 shaker (Zhicheng Analytical Instrument Manufacturing Co., Shanghai, China), and then the solution was filtered. $NH_4^+-N$ and $NO_3^--N$ contents were measured using an Auto Analyzer 3 Continuous Flow Chemical Analyzer (Bran+Luebbe, Norderstedt, Germany), according to the suggested method [34].

### 2.3.3. Total Soil N

A total of 21 g air-dried soil passed through a 100 mesh sieve to measure TN content according to semi-micro Kelvin method [20].

$$TN (g/kg) = (V - V_0) \times 14 \times 10^{-3} \times C \times 1000/m, \tag{2}$$

where V is volume of acid standard solution that is used in titration of test solution (mL), $V_0$ is volume of acid standard solution used in titration of blank (mL), c is 0.01 mol/L (1/2 $H_2SO_4$), 14 is molar mass of N atom (G/mol), $10^{-3}$ converts mL to L, and m is weight of air-dried soil sample (g).

### 2.3.4. Thousand Grain Weight, Yield and Harvest Index

Yields were harvested and measured at physiological maturity from a total of 2 $m^2$ area (about 11 plants) from each plot to measure yield, and were weighed after naturally dried. HI was calculated by,

$$HI = Grain\ yield\ (kg/ha)/above\ ground\ biomass\ (kg/ha) \tag{3}$$

### 2.4. Statistical Analyses

All graphs were drawn with SigmaPlot 12.5 (SPSS Inc., Chicago, IL, USA), Origin 8 (OriginLab Corporation, Northampton, MA, USA) and R 4.0.2 (R Core Team, https://www.r-project.org/, accessed on 11 December 2020, Auckland, New Zealand), and all data were analyzed with SPSS Statistics 21.0 (IBM Inc., Chicago, IL, USA). The means of different treatments were considered significant with least significant difference test at $p < 0.05$.

## 3. Results

### 3.1. N Accumulation and Distribution in Maize Plant

The total nitrogen content and their distribution in different plant organs at flowering and maturity stages in spring and autumn cropping season during 2018–2019 were mea-

sured and presented in Figure 2. The supplementary irrigation and nitrogen increment have significant effects on the TN content during all plant stages and growing seasons. For example, for spring maize in 2018, total plant N contents of rainfed and supplementary irrigation were 0.65% and 0.62% (stem) and 1.99% and 1.92% (leaf), respectively ($p < 0.05$). The TN contents of stems, leaves, and bract+cob increased first and then decreased with increment of nitrogen amounts. The TN of plant content of nitrogen application was significantly higher than that of no nitrogen application ($p < 0.05$). For autumn maize in 2018, the TN content of bract+cob under irrigation increased by 4.7% compared with rainfed. Grains TN contents under N0, N150, N200, N250, and N300 were 1.6%, 1.8%, 2.0%, 2.2%, and 2.4% at mature stage, respectively, and under N300 was approximately 1.5-fold that of under N0. The TN contents of stems and leaves increased with increase in N amounts.

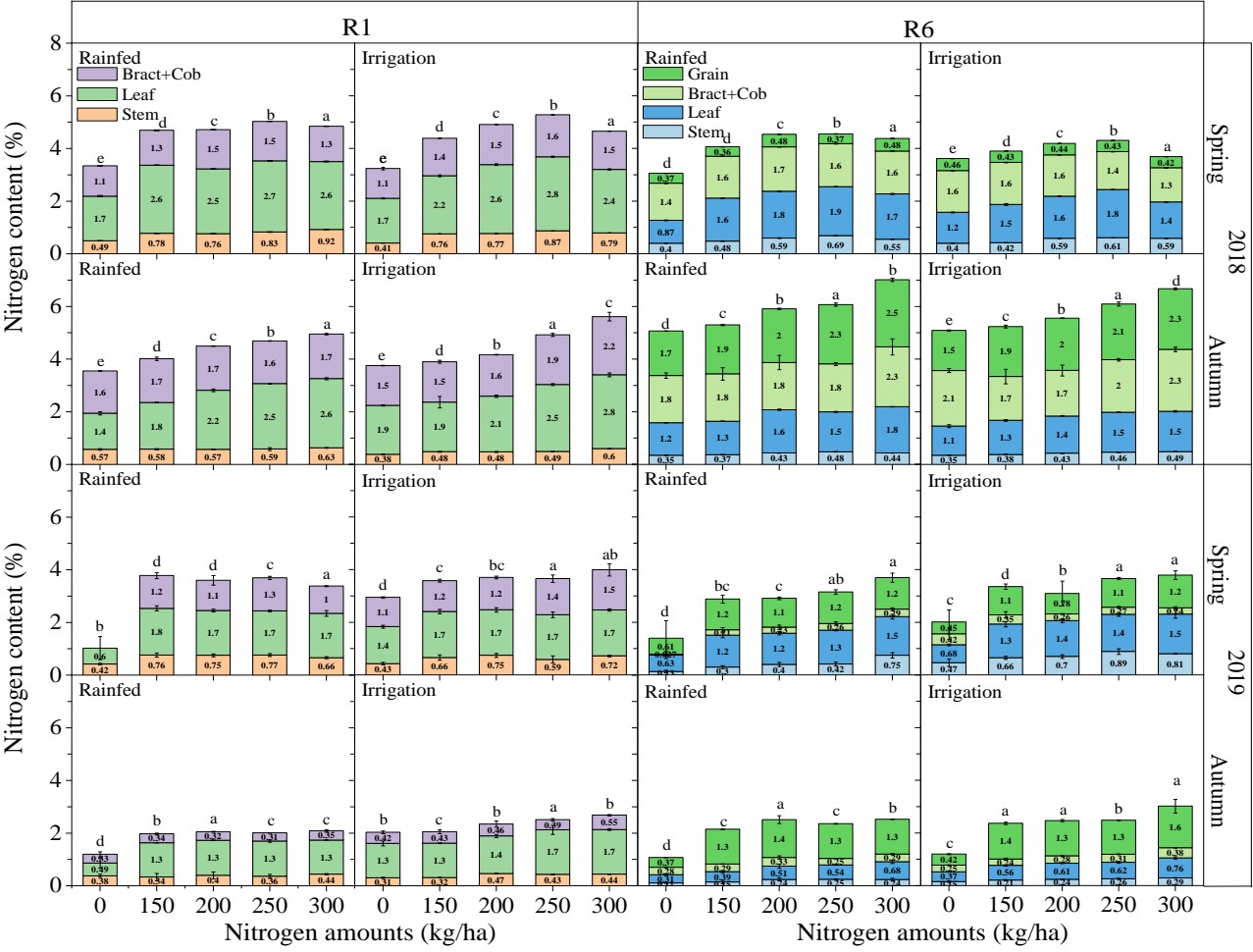

**Figure 2.** The total nitrogen content of maize under different water conditions and nitrogen amounts in 2018–2019. Different lowercase letters indicated that there were significant differences ($p < 0.05$); the bars are the SE ($n = 3$). R1 stands for flowering stage and R6 stands for maturity stage of maize. While, the values inside the bar graph are percent values of three replicates.

For spring maize in 2019, the TN contents of leaves and bract+cob of irrigation increased by 11.8% and 36.9% compared with rainfed at flowering stage, respectively. The TN contents of stems in rainfed and irrigated were 0.4% and 0.7% at mature stage, respectively. Grain TN content of nitrogen application was significantly higher than that of no nitrogen application ($p < 0.05$). For autumn maize in 2019, the TN contents of stems, leaves, and bract+cob at flowering stage were higher than that at mature stage, and that of grains under irrigation was 3.8% higher than that under rainfed. Grain TN contents of

N0, N150, N200, N250, and N300 were 0.39%, 1.30%, 1.34%, 1.39%, and 1.45%, respectively. Grain TN content under N300 was fourfold under N0.

The results of two-year experiment showed that water condition and nitrogen application affected TN content of maize plant. Pl TN content of nitrogen application was significantly higher than that of no nitrogen application ($p < 0.05$). The average TN contents of all organs in spring maize were higher than those of autumn maize (Figure 2).

### 3.2. Soil Inorganic N

The soil $NO_3^-$-N and $NH_4^+$-N contents covering two cropping seasons during 2018 and 2019 are shown in Figure 3. The pooled data analysis of $NO_3^-$-N and $NH_4^+$-N under the irrigation regime and nitrogen level were significantly higher in spring season than in autumn season ($p < 0.05$). There was also a significant effect of irrigation regime and nitrogen level in both seasons. For instance, under rainfed and irrigation conditions, the $NO_3^-$-N and $NH_4^+$-N contents were relatively high under high nitrogen treatment, the orders were both N250 > N300 > N200 > N100. However, as whole, soil inorganic contents were higher under rainfed condition than supplementary irrigation condition. Across 2018–2019, the $NO_3^-$-N and $NH_4^+$-N showed a tendency that increased first and then decreased with increasing nitrogen amounts, thereby reaching maximum under N250. Average $NO_3^-$-N contents of rainfed and supplementary irrigation were 9.2 and 8.4 g/kg, respectively, which shows 9.5% significant difference between the two irrigation treatments ($p < 0.05$). Taking average over the two year pooled data, $NH_4^+$-N contents of treatments N0, N150, N200, N250, and N300 were 3.8, 6.2, 9.6, 11.2, and 13.2 g/kg, respectively. The $NO_3^-$-N content of N250 was significantly higher than those of other nitrogen levels ($p < 0.05$). Average $NH_4^+$-N contents of rainfed and supplementary irrigation were 16.7 and 15.5 g/kg, respectively. Average $NH_4^+$-N content under rainfed was significantly higher than supplementary irrigation ($p < 0.05$). Average $NH_4^+$-N contents of N0, N150, N200, N250, and N300 were 12.1, 13.7, 15.4, 22.4, and 16.8 g/kg, respectively; the order was N250 > N300 > N200 > N150 > N0.

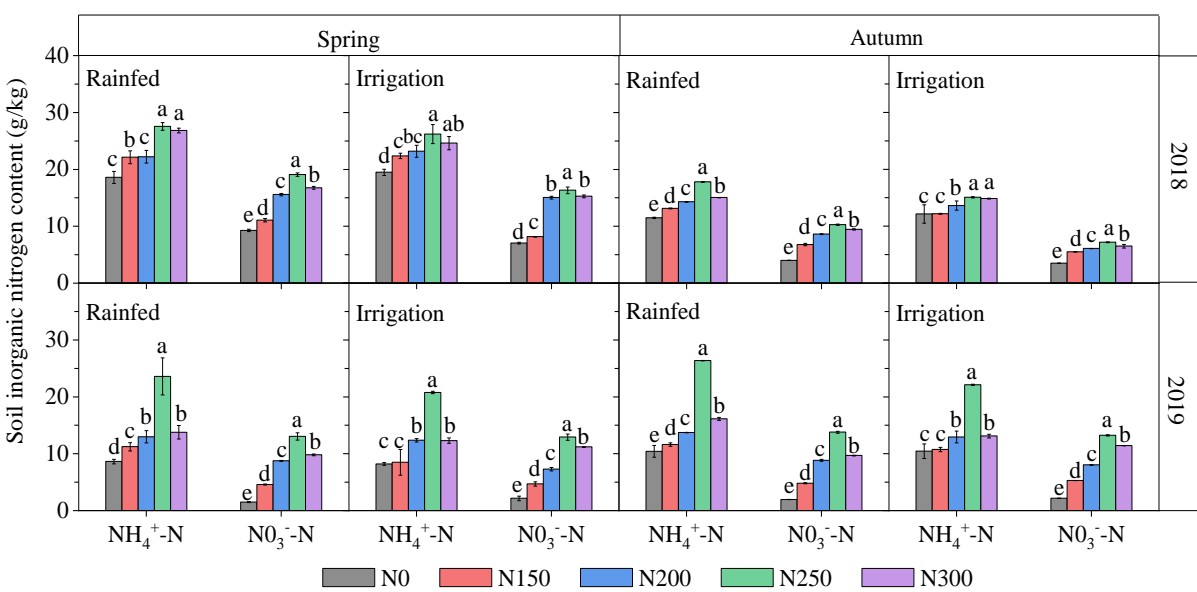

**Figure 3.** The soil inorganic nitrogen content under different water conditions and nitrogen amounts at maize maturity stage in 2018–2019. Different lowercase letters indicated that there were significant differences ($p < 0.05$); the bars are the SE ($n = 3$).

### 3.3. Soil Total Nitrogen (TN)

Figure 4 presents the soil total nitrogen content under two water regimes and five nitrogen treatments during the spring and autumn seasons over 2018 and 2019. The irri-

gation and nitrogen dependent response was found, such as, TN content under rainfed was higher than supplementary irrigation throughout the experimental duration. For example, TN content under rainfed and supplementary water treatment in spring season were 1.51 and 1.50 g/kg and in autumn season were 2.44 and 2.20 g/kg, respectively, during 2018 and were 2.15 and 2.12 (spring maize) and 2.82 and 2.68 g/kg (autumn maize), respectively, during 2019. Similarly, nitrogen increment significantly increased (N250 > N300 > N200 > N150 > N0) the TN content, however, the highest nitrogen treatment (N300) reduced the TN content compared to N250 in one-year double cropping maize during 2018. The same trend was also found in spring season maize during 2019.

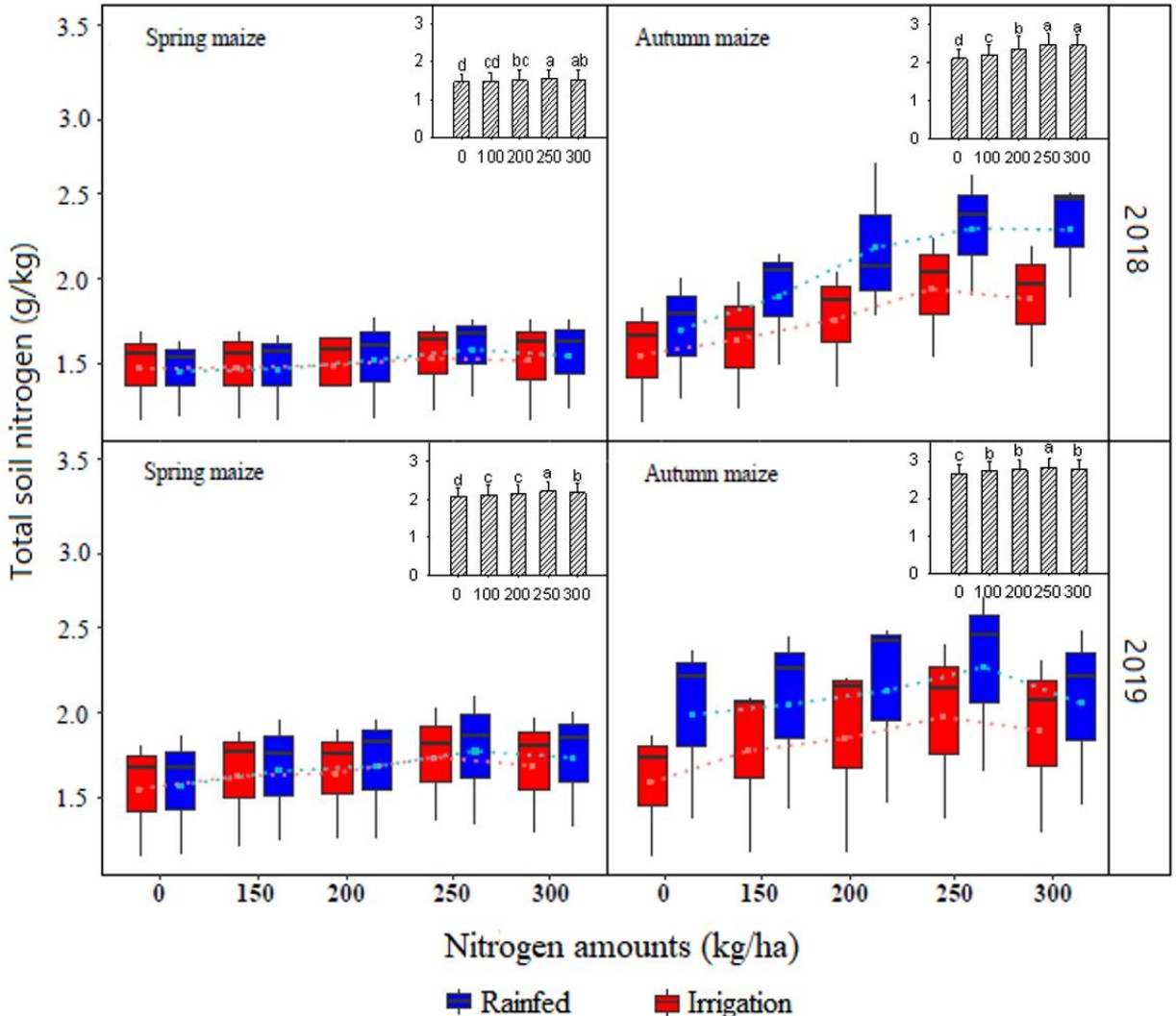

**Figure 4.** The total soil nitrogen content under different water conditions and nitrogen amounts in 2018–2019. The points inside the box plot represents the average of three repeated data. The small picture represents the total soil nitrogen contents under different nitrogen amounts; the bars are the SE. Values followed by different small letters are significantly different (*p* < 0.05).

Taking all together, the two-year results showed that TN contents increased first and then decreased with increase of nitrogen amounts. The TN content of rainfed was higher than that of supplementary irrigation, and autumn maize was higher than that of spring maize.

### 3.4. Soil Organic Carbon (SOC) and Microbial Biomass Carbon (MBC)

The contents of soil organic carbon and microbial biomass carbon during 2018–2019 are shown in Table 1. The results showed that the MBC content of rainfed was significantly higher than that of supplementary irrigation, while there was no significant difference in SOC. The MBC and SOC contents increased with the enhance of nitrogen application. For example, in 2018, the SOC and MBC of spring maize were 16.57 and 0.31 g/kg (supplementary irrigation) and 14.00 and 0.12 g/kg (rainfed), respectively. The soil organic carbon contents of N0, N150, N200, N250 and N300 in autumn maize in 2018 were 14.68, 16.36, 16.66, 15.89 and 15.13 g/kg, respectively. The MBC content was 0.07, 0.12, 0.13, 0.18 and 0.31 g/kg, respectively. The same trend is expected during 2019.

**Table 1.** The soil organic carbon and microbial biomass carbon in maize field at R6 under different water conditions and nitrogen amounts in 2018–2019.

| Treatment | | Soil Organic Carbon (g/kg) | | | | Microbial Biomass Carbon (g/kg) | | | |
|---|---|---|---|---|---|---|---|---|---|
| | | 2018 Spring | 2018 Autumn | 2019 Spring | 2019 Autumn | 2018 Spring | 2018 Autumn | 2019 Spring | 2019 Autumn |
| Rainfed | N0 | 19.88 d | 14.17 c | 12.50 bc | 10.89 c | 0.29 b | 0.07 d | 0.11 d | 0.08 d |
| | N150 | 18.28 b | 16.08 b | 11.99 c | 11.88 b | 0.31 b | 0.15 c | 0.15 c | 0.12 c |
| | N200 | 16.95 c | 16.28 a | 12.72 bc | 12.38 b | 0.35 a | 0.16 bc | 0.16 b | 0.13 bc |
| | N250 | 16.61 c | 16.29 a | 12.85 ab | 13.7 a | 0.35 a | 0.19 b | 0.18 ab | 0.14 b |
| | N300 | 11.14 a | 16.33 a | 13.73 a | 14.36 a | 0.27 c | 0.27 a | 0.18 a | 0.17 a |
| Irrigation | N0 | 10.36 c | 15.19 d | 12.6 b | 11.22 d | 0.10 b | 0.08 c | 0.10 c | 0.11 c |
| | N150 | 10.26 c | 16.64 b | 12.87 ab | 12.21 c | 0.11 b | 0.08 c | 0.14 b | 0.12 c |
| | N200 | 15.36 b | 17.04 a | 12.98 ab | 13.04 bc | 0.10 b | 0.10 c | 0.15 ab | 0.14 b |
| | N250 | 16.6 a | 15.49 c | 13.66 a | 13.86 ab | 0.15 a | 0.16 b | 0.16 a | 0.16 a |
| | N300 | 17.42 a | 13.93 e | 13.79 a | 14.69 a | 0.15 a | 0.34 a | 0.16 a | 0.18 a |
| *p* value | | | | | | | | | |
| Water | | 0.0293 | 0.0092 | 0.2213 | 0.0625 | 0.0001 | 0.2053 | 0.0008 | 0.2519 |
| Nitrogen | | 0.0001 | 0.0001 | 0.118 | 0.0001 | 0.0001 | 0.0001 | 0.0001 | 0.0001 |
| Water × Nitrogen | | 0.0001 | 0.0001 | 0.668 | 0.9832 | 0.0001 | 0.0001 | 0.7469 | 0.2504 |

Values followed by different small letters within a column are significantly difference ($p < 0.05$, $n = 3$).

Two years of results showed that the SOC content of autumn maize was higher than that of spring maize, while MBC content showed an opposite result (Table 1). The MBC content under nitrogen application was significantly higher than that under no nitrogen application ($p < 0.05$). The MBC contents of supplementary irrigation and rainfed were significantly different ($p < 0.05$), and the values were 0.14 and 0.20 g/kg, respectively. The SOC content showed no significant difference between rainfed and supplementary irrigation. The SOC contents of N0, N150, N200, N250, and N300 were 13.35, 13.77, 14.59, 14.88, and 14.42 g/kg, respectively.

### 3.5. Thousand Grains Weight (TGW), Yield, and Harvest Index (HI)

Results indicated that TGW, grain yield, and HI rose significantly under the supplementary irrigation condition and decreased under the rainfed condition in 2018 and 2019 seasons (Table 2). The increase in these parameters under the supplementary irrigation over the rainfed irrigation was 2.5% (TGW), 35.8% (Yield), and 28.0% (HI) in spring 2018 and 2.4% (TGW), 10.1% (Yield), and 5% (HI) in spring 2019. Similarly, the TGW and Yield of maize improved with increase of nitrogen application rate during the spring season of 2018 and 2019. The same increasing trend under the nitrogen application was also noted during the autumn season of both years with highest positive effect was observed in treatment N300. For instance, the TGW of maize during the autumn season 2018 of treatment N300 was 38.2%, 7.7%, 4.1%, and 3.2% higher over N0, N150, N200, and N250, respectively. During the same season the average yield of maize increased with the increas-

ing order of nitrogen rate such as 3448 (N0), 4654 (N150), 6195 (N200), 6798 (N250), and 7011 (N300) kg/ha. We also estimated the percent increase yield of maize and found that the Yield under N300 was 103.3% and 45.0% higher than treatment N0 during 2018 and 2019, respectively.

The two-year average results showed that TGW, Yield, and HI of supplementary irrigation were 7.7%, 22.0%, and 15.2% higher than those of rainfed condition, respectively. Yields of N150, N200, N250, and N300 were 37.1%, 56.0%, 64.3%, and 70.0% higher than that of N0 condition, respectively. Average yield of spring maize was higher than that of autumn maize, and the values were 6090 and 5621 (2018) and 6676 and 4675 kg/ha (2019), respectively.

**Table 2.** The thousand grains weight, yield and harvest index of maize under different water conditions and nitrogen amounts in 2018–2019.

| Treatment | | TGW (g) | | | | Yield (kg/ha) | | | | HI | | | |
|---|---|---|---|---|---|---|---|---|---|---|---|---|---|
| | | 2018 Spring | 2018 Autumn | 2019 Spring | 2019 Autumn | 2018 Spring | 2018 Autumn | 2019 Spring | 2019 Autumn | 2018 Spring | 2018 Autumn | 2019 Spring | 2019 Autumn |
| Rainfed | N0 | 227.8 c | 223.9 d | 133.1 c | 199.0 a | 3651 e | 4135 d | 4756 c | 2909 b | 0.21 c | 0.47 a | 0.41 ab | 0.33 a |
| | N150 | 289.7 b | 278.4 bc | 151.8 b | 194.4 a | 4778 d | 4120 d | 6432 b | 3053 b | 0.24 b | 0.41 c | 0.48 a | 0.22 b |
| | N200 | 293.6 b | 275.1 c | 150.3 b | 204.3 a | 5415 c | 6207 c | 6341 b | 3279 ab | 0.28 b | 0.42 b | 0.41 b | 0.22 b |
| | N250 | 304.1 a | 280.8 b | 150.3 b | 204.7 a | 5677 b | 6757 b | 6652 b | 4024 a | 0.27 a | 0.40 c | 0.31 c | 0.23 b |
| | N300 | 293.6 b | 297.6 a | 156.4 a | 206.4 a | 6304 a | 7219 a | 7599 a | 4002 a | 0.27 a | 0.42 b | 0.39 b | 0.2 b |
| Irrigation | N0 | 241.8 d | 206.6 c | 142.7 b | 206.8 d | 5554 d | 2761 d | 4500 c | 4373 c | 0.27 c | 0.42 d | 0.44 ab | 0.28 a |
| | N150 | 294.5 c | 274.4 b | 155.2 a | 241.6 bc | 6780 c | 5187 c | 7250 b | 5513 b | 0.34 b | 0.56 a | 0.46 a | 0.34 a |
| | N200 | 305.6 a | 296.8 a | 154.4 a | 235.1 c | 7242 b | 6184 b | 7357 b | 6513 a | 0.34 a | 0.52 b | 0.37 b | 0.34 a |
| | N250 | 299.5 b | 296.2 a | 154.9 a | 257.1 ab | 7806 a | 6839 a | 7740 ab | 6519 a | 0.34 a | 0.47 c | 0.43 ab | 0.28 a |
| | N300 | 302.9 ab | 295.6 a | 152.7 a | 267.6 a | 7692 a | 6804 a | 8131 a | 6561 a | 0.32 a | 0.43 d | 0.42 ab | 0.29 a |
| | | | | | | *p* value | | | | | | | |
| Water | | 0.0132 | 0.0750 | 0.0002 | 0.0001 | 0.0003 | 0.0151 | 0.0027 | 0.0018 | 0.0020 | 0.0001 | 0.2905 | 0.0381 |
| Nitrogen | | 0.0001 | 0.0001 | 0.0001 | 0.0001 | 0.0001 | 0.0001 | 0.0001 | 0.0004 | 0.0001 | 0.0001 | 0.0531 | 0.1118 |
| Water × Nitrogen | | 0.0013 | 0.0001 | 0.0024 | 0.0011 | 0.0001 | 0.0001 | 0.1106 | 0.1613 | 0.0067 | 0.0001 | 0.1511 | 0.0132 |

Values followed by different small letters within a column are significantly different ($p < 0.05$, $n = 3$).

### 3.6. Correlation between Plant N, Soil C–N, and Yield

The relationship among nitrogen rate, MCB, SOC, $NO_3^-$-N, $NH_4^+$-N, stem nitrogen (Stem-N), leaf nitrogen (Leaf-N), grain nitrogen (Grain-N), TGW, HI and yield were analyzed using regression correlation coefficient (Figure 5). The results of 2-year experiment showed that nitrogen application had a significantly positive correlation with MBC, $NO_3^-$-N, $NH_4^+$-N, Stem-N, Leaf-N, Grain-N, TGW, and yield ($p < 0.01$), reveals that nitrogen application is the prerequisite for higher yield of maize. The MBC was significantly positively correlated with SOC, $NO_3^-$-N, $NH_4^+$-N, Stem-N, Leaf-N, bract+cob−N, and TGW ($p < 0.01$) and had a significantly negative correlation with TN ($r = -0.30$) and HI ($r = 0.26$, $p < 0.01$). The SOC showed an extremely significant positive correlation with $NO_3^-$-N and $NH_4^+$-N ($p < 0.01$), and significantly negative correlation with TN ($p < 0.01$). The $NO_3^-$-N content had an extremely significant positive correlation with $NH_4^+$-N, Stem-N and Leaf-N contents, TGW, and yield ($p < 0.01$) but had significantly negative correlation with TN, Grain-N content, and HI ($p < 0.001$). The $NH_4^+$-N content was significantly negative correlated with TN, Grain-N, and HI ($p < 0.001$) and extremely significantly positively correlated with Stem-N and Leaf-N contents, TGW, and yield ($p < 0.001$). The TN content had a significantly negative correlation with Stem-N, Leaf-N, bract+cob–N, TGW, and yield ($p < 0.001$) but had an extremely significantly positive correlation with Grain-N ($r = 0.53$, $p < 0.001$). Stem-N and Leaf-N contents had an extremely significant positive correlation with yield ($p < 0.001$), with $r$ values of 0.77 and 0.65, respectively. Grain-N content had an extremely significantly positive correlation with HI ($r = 0.51$, $p < 0.001$) and a significantly positive correlation between HI and yield ($r = 0.35$, $p < 0.001$).

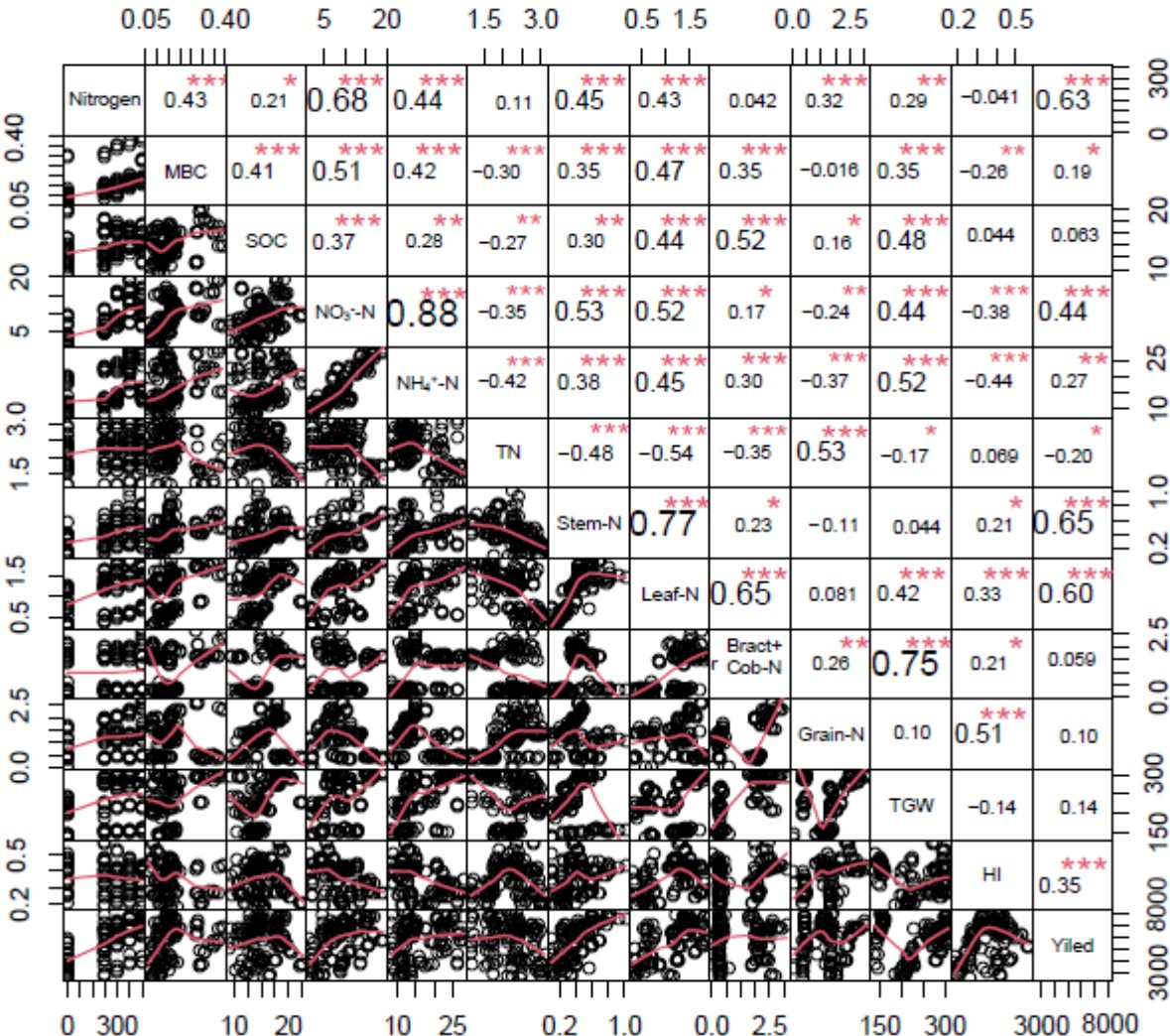

**Figure 5.** Regression of nitrogen and soil carbon–nitrogen, plant nitrogen content, thousand grains weight (TGW), harvest index (HI) and yield in 2018–2019. Stem-N, Leaf-N, Bract+Cob-N, and Grain-N represent the nitrogen content (%) of stem, leaf, cob + bract and grain, respectively. *, **, and *** represent significant levels of correlation coefficients of $p < 0.05$, $p < 0.01$, and $p < 0.001$, respectively.

## 4. Discussion

### 4.1. Water and Nitrogen in Relation to Plant Nutrients

N and water are the main two limiting factors for maize growth. Plants mainly rely on N present in the soil to fulfill the nutrient requirement [35,36]. Nitrogen application promotes plant growth, increases plant N content, and proportion of N that is allocated to grain [37]. Excess nitrogen fertilizer affects quality of seeds [38,39]. When amount of nitrogen application exceeds a certain value, the biological and economic yield of maize do not increase. Water and fertilizer interaction can effectively promote uptake and transportation of nutrients to the plant tissues [40,41]. In present study, we found high accumulation of N in the vegetative plant parts (stems and leaves) from seedling to flowering stage. While, during the reproductive stage the TN contents of stems, leaves, and bract + cob was gradually transferred to grain. After flowering, it is because N absorption of maize significantly increases with increasing application of nitrogen fertilizer (Figure 2).

The results of this experiment showed that nitrogen applied alone or in combination with supplementary irrigation can increase N contents of maize plant during flowering period. N contents of stems and leaves were significantly positively correlated with yield. N accumulation increased during the vegetative growth period, which was conducive

to transfer to grain in repining stage, which improved TGW and yield. Supplementary irrigation was beneficial to N accumulation in grains, thereby indicating that suitable soil moisture conditions improved N transport and distribution. Considering difference in climatic conditions between spring and autumn, low temperature environment in late growth stage of autumn maize affected N absorption and transportation, thereby making N contents of various organs lower than those of spring maize.

### 4.2. Impacts of Water and Nitrogen Application on Soil Nitrogen

Plants prefer to more readily absorb the inorganic N ($NO_3^-$-N and $NH_4^+$-N) than organic N, while most N in soil is organic form. Therefore, a continuous nitrification in soil is required to convert the soil organic nitrogen into inorganic form. Liu et al. [42] mentioned that nitrification in soil is most vigorous when soil water content is 50–60% of maximum field capacity of soil, and it can be inhibited to a certain extent in the presence of too high or low soil moisture content. Excess water and fertilizer not only reduce the nitrification process but also cause a large amount of $NO_3^-$-N leaching and groundwater pollution [43,44]. The main factor of $NO_3^-$-N residue in the soil is nitrogen application [45], which can move down due to excessive water supply [46]. In this study, the results of two-year experiment showed no significant difference in $NO_3^-$-N contents in the soil between rainfed and supplementary irrigation, this might be because of leaching down of $NO_3^-$-N due to the heavy rainy season in subtropical region.

Furthermore, in this current study, irrigation amount and fertilization rate had prominent effects on soil $NO_3^-$-N content (Figure 3). For instance, under the nitrogen application treatment, the $NO_3^-$-N and $NH_4^+$-N contents were first increased and then decreased with increase in nitrogen amounts while $NO_3^-$-N content reduced under the supplementary irrigation. This might mean $NO_3^-$-N is very soluble in water and moves with water [47]. In addition, it was previously reported that mineralization of organic N was affected by seasons [48], in line with this we found seasonal dependent $NO_3^-$-N content such as spring maize was higher than that of autumn maize. The possible reason might be lower rainfall in spring season than autumn season during 2018–2019 in the region (Figure 1). Nitrogen application was significantly positively correlated with soil $NO_3^-$-N and $NH_4^+$-N contents; the $NO_3^-$-N and $NH_4^+$-N contents were significantly positive correlated to yield and TGW, thereby indicating that nitrogen application in a certain range can increase economic yield. When nitrogen amount is 100 kg/ha, the $NO_3^-$-N content in 0–5 cm soil layer could increase by 31–45% compared with no nitrogen application [49,50]. The highest nitrogen treatment (N300) led to reduced TN content compared to N250 in one-year double cropping maize. The possible cause might be that maize grow well and have poor ventilation in N300 resulting in a higher surface temperature, which accelerated the mineralization rate of soil TN content, and increased nitrogen uptake by plants, resulting in a lower soil total nitrogen [51].

### 4.3. Soil Carbon Pool

The MBC is an important part of soil carbon pool. Climatic conditions and cultivation measures can affect the amount of MBC [42,52]. The results of present study showed that MBC under nitrogen application treatment was significantly higher, relative to without nitrogen application treatment. The MBC was significantly positively correlated with yield. The SOC reached its peak when nitrogen application 250 kg/ha was applied, and water conditions had no significant effect on SOC. Rainfed combined with nitrogen application was beneficial to MBC accumulation, and MBC of spring maize was higher relative to autumn maize, mainly because climatic conditions during the maturity stage of spring maize were conducive to survival of microorganisms (Table 1).

### 4.4. The Relationship between Water, Fertilizer and Yield

Water and nitrogen application can significantly affect TGW and yield of maize. Similarly, HI has a positive linear correlation with yield of the crop [53]. Our result



showed that the TGW and yield increased with the increase of nitrogen application rate and supplementary irrigation improved HI (Table 2). It is reported that optimum soil water condition and N amount can improve utilization efficiency of nitrogen fertilizer, and integration of water and fertilizer can significantly increase yield [54–56]. On basis of obtaining a certain biological yield, increasing the HI was key to enhance crop production. This result is in line with previous reports that HI of different crop cultivars was significantly correlated with grain yield of wheat and canola [57,58]. Our research showed that there were overall increases in TGW, HI and yield under the supplementary irrigation and nitrogen application treatments (Figure 5).

## 5. Conclusions

This research clarified effects of water conditions and nitrogen amounts on maize yield, plant N uptake and soil C–N conversion and provided reliable technical support for water and fertilizer saving of maize in double cropping in subtropical regions.

N250 combined with supplementary irrigation can increase soil inorganic N components, TN content, and SOC content and improve soil fertility. In addition, yield of autumn maize was lower than that of spring maize. Considering the goal of reducing the impact on water, and also the reduced fertilizer costs as well as decreasing the risk of groundwater pollution while maintaining high maize yield, a drip irrigation along with N fertilization rate of 250 kg/ha under the phased drought condition, was recommended to obtain maximum grain nitrogen accumulation and maize grain yield simultaneously in the subtropical Guangxi region. This research is of great significance for the area, however, to measure the WUE, NUE. This result was beneficial to maize N absorption and transfer, promoted maize growth, and increased yield. The method to improve farming and enhance annual yield of subtropical maize requires further research.

**Author Contributions:** Conceptualization, G.-Y.W., Y.-X.H., Y.-X.L., S.A. and X.-B.Z.; data curation, G.-Y.W.; formal analysis, Y.-X.H.; investigation, S.A.; methodology, Y.-X.L.; project administration, X.-B.Z.; software, G.-Y.W.; supervision, X.-B.Z.; Writing—original draft, Y.-X.H.; Writing—review and editing, S.A. and X.-B.Z. All authors have read and agreed to the published version of the manuscript.

**Funding:** The research was funded by the National Natural Science Foundation of China (31760354) and Natural Science Foundation of Guangxi Province (2019GXNSFAA185028).

**Institutional Review Board Statement:** Not applicable.

**Informed Consent Statement:** Not applicable.

**Data Availability Statement:** The data that support the findings of this study are available from the corresponding author upon reasonable request.

**Conflicts of Interest:** The authors declare no conflict of interest.

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
