# Peer review of "Effects of Supplement Irrigation and Nitrogen Application Levels on Soil Carbon–Nitrogen Content and Yield of One-Year Double Cropping Maize in Subtropical Region"

_water, doi:10.3390/w13091180_

Round 1

Reviewer 1 Report

It would  be interesting to have water analysis and total amount or consumption per crop and year. It´s important when it has levels of NO3 or another important ions.

Out of curiosity, I would know the pH of soil.

In the line 277 of manuscript you have a mistake, because you have reference to table nº 1 and it would say table 2. 

Author Response

Dear professor,

We greatly appreciate the thorough and thoughtful comments provided on our submitted article. We made sure that each one of the reviewer comments has been addressed carefully and the paper is revised accordingly.

Author's Reply to the Review Report (Reviewer 1)

Comment: Out of curiosity, I would know the pH of soil.

√Line 100: We had added pH of based on the soil fertility, but we don’t measure the pH of soil in 2018-2019.

Comment: Line 277: you have a mistake, because you have reference to table nº 1 and it would say table 2.

√Line 277: We had checked and revised it.

Reviewer 2 Report

Recommendations to authors

  • Line 21: "combined"?
  • Line 31-91: Present language quality is not good enough and needs significantly to be improved. It is difficult to understand the intended meaning of some of the sentences. In some parts of the text, the language errors distract from the overall quality of the work. Whilst the content is clear, the standard of English needs to be improved throughout the manuscript and it would benefit from a thorough proofread by a native English speaker to improve grammar. Please check the use of English throughout the manuscript.
  • Line 34-36: Please rephrase; it makes no sense. Do you mean "water-intensive" instead of " intensive food"?
  • Line 38-41: Please rephrase. The present language quality is not good enough and needs significantly to be improved.
  • Line 79: Do you mean "alone" instead of "aione" and "combined" instead of "combine"?
  • Line 81-82: Please rephrase.
  • Line 157-159: Please replace "V0" with "V0" (make the zero subscript).
  • Line 172, 180,183,192, 199, 203, 211, 252 (and throughout the whole text): Please change case; this is the “p-value”, so you must replace “P<0.005” with “p<0.05”.
  • Line 242. Replace "than" with "compared to".
  • Line 243. Not enough explanation/reasoning why the highest nitrogen treatment (N300) lead to reduced TN content compared to N250 in one-year double cropping maize during both 2018 and 2019.
  • Line 336: Please consider replacing "alone and combine application of nitrogen" with "nitrogen applied alone or in combination with supplementary irrigation", or "combined use of irrigation and nitrogen", or "effects of irrigation and nitrogen application".
  • Line 356: Please add "in the soil".
  • Line 359-366. Please rephrase.
  • Line 393: Please consider adding a limitations section for the proposed N fertilization rate of 250 kg ha-1 (the one recommended to obtain maximum grain nitrogen accumulation and maize grain yield)
  • Line 399-403. Please rephrase the sentence. Also: “Considering the goal of conserving both water and fertilizer". Suggestion: "Considering the goal of reducing the impact on water, and also the reduced fertilizer costs".

Author Response

Dear professor,

We greatly appreciate the thorough and thoughtful comments provided on our submitted article. We made sure that each one of the reviewer comments has been addressed carefully and the paper is revised accordingly.

please find in attachment.

Reviewer 3 Report

The paper ‘Effects of Supplement Irrigation and Nitrogen Application Levels on Soil Carbon–Nitrogen Content and Yield of One-Year Double Cropping Maize in Subtropical Region’ presents results of two year field experiments of cultivation under specific conditions including five levels of nitrogen inputs and two types of water regimes (rainfall and additional? Irrigation).

The results are presented clearly.

There are some questions concerning interpretation of the data

  • The Harvest Index is not sufficiently well exploited, in my opinion.

389-390 ‘On basis of obtaining a certain biological yield, increasing the HI was key to enhance crop production.’ Looking at Table 2, which in fact do not show a combination of two types of watering and levels of N fertilization, it is visible than HI is not highest for the higher values of N input. It seems that N150 gives better results than N250. Of course the yields are highest for the highest level of N fertilization (N300).

  • The part concerning TGW, yields and HI should be better presented including results and discussion
  • I agree that the subject needs more studies and even the recommendation of N250 is better than using as high level of N input as possible, noting that the increase of yields grow together with increasing N input.
  • Some details are marked in pdf in lines:

24 – 25:              Yield and TGW N300 were 287 kg ha-1 and 3.1 g higher than those of N250, and N250 were 59.4% and 23.1% higher than those of N0, respectively.     --> that sentence is very unclear

41:  ‘very’ --> ‘more’

81: To best of our knowledge there is merely

96: temperate climate --> subtropical climate (Is the climate temperate or subtropical? The annual average air temperature of 21.6 C is really high. In temperate Europe it is around 10 and less

129-130: 100 mesh sieve --> I would add here in brackets (0.149 mm) and it is not necessary to give that info again

138: 100-mesh – use consequently with ‘–‘ or without

146: SOC was determined

155: here ‘(<0.15 mm)’ can be deleted if mentioned above, in line 130

156: semi-micro Kelvin method is not given in publication [20], I did not check the other sources but it is making reader rather suspicious

199-200: The TN contents of all organs in spring maize were higher than those of autumn maize. --> The average TN contents? I can see something opposite in the Fig. 2 for year 2018. The bars are higher for Autumn than for Spring. Have you compared some averages for all Spring and all Autumn results?

213-214: contents were higher under the treatment N250 followed N300, and then N200 over the 2018 and 2019. --> It is very unclear.

258: spring maize and were 16.57 --> delete ‘and’

259-260: The soil organic carbon

275: and HI of rose --> delete ‘of’

277: Table 1 --> Table 2

292: Yields of N300, N250, N200, and N150 were 37.1%, 56.0%, 64.3%, and 70.0% higher --> some changing of order necessary?

357: this is might be because --> delete ‘is’

384-385: Now Table 2 is not mentioned in the text. Treatment: for Rainfed and Irrigation are there some averages of all Ns? Is it possible to present the data in similar way as in Table 1?

386- 392: In my opinion that part should be better discussed. It is well visible that ‘Yield gradually increased with increase in nitrogen amounts’ even more in present Table 2 than in figure 5 and there is a question why harvest index is important? It is written that ‘On basis of obtaining a certain biological yield, increasing the HI was key to enhance crop production.’ But why? And how that index is reflected in your data?

Overall statement: The paper presents very valuable data but the point of ‘biological yield’ and importance of increasing HI could be better described.

Author Response

Dear professor,

We greatly appreciate the thorough and thoughtful comments provided on our submitted article. We made sure that each one of the reviewer comments has been addressed carefully and the paper is revised accordingly.

please find it in attachment.
